# Bladder Ultrastructure and Urinary Cytokine Abnormality in Patients with Recurrent Urinary Tract Infection and the Changes after Intravesical Platelet-Rich Plasma Injections

**DOI:** 10.3390/biomedicines10020245

**Published:** 2022-01-24

**Authors:** Jia-Fong Jhang, Han-Chen Ho, Yuan-Hsiang Hsu, Yuan-Hong Jiang, Hann-Chorng Kuo

**Affiliations:** 1Department of Urology, Hualien Tzu Chi Hospital, Buddhist Tzu Chi Medical Foundation, 707, Section 3, Chung-Yang Road, Hualien 97004, Taiwan; alur1984@hotmail.com (J.-F.J.); redeemerhd@gmail.com (Y.-H.J.); 2Department of Urology, School of Medicine, Tzu Chi University, Hualien 97004, Taiwan; 3Institute of Medical Sciences, Tzu Chi University, Hualien 97004, Taiwan; 4Department of Anatomy, Tzu Chi University, Hualien 97004, Taiwan; hcho@gms.tcu.edu.tw; 5Department of Pathology, Hualien Tzu Chi Hospital, Buddhist Tzu Chi Medical Foundation, Hualien 97004, Taiwan; yhhsu@mail.tcu.edu.tw; 6Department of Pathology, School of Medicine, Tzu Chi University, Hualien 97004, Taiwan

**Keywords:** inflammation, urothelium, recurrent urinary tract infection, platelet-rich plasma, regeneration

## Abstract

This study investigates the bladder from patients with recurrent urinary tract infection (rUTI) at baseline and after intravesical platelet-rich plasma (PRP) injections. Patients with rUTI who underwent repeated intravesical PRP injections provided bladder and urine specimens at baseline and after treatment. Bladder specimens were investigated with electron microscopy and Western blotting. The urine sample was analyzed with commercially available Milliplex immunoassays. A total of 29 patients were enrolled. At baseline, the rUTI bladders exhibited defects of integrity in umbrella cells, a widened tight junction, and lysed organelles. Intracellular bacterial community incubations in the epithelial cells were also noted. Improvement in bladder defects after PRP injection was noted in 25–42% of patients. Bladder UPK3 expression was significantly lower in the patients with rUTI than in controls. Baseline levels of urinary inflammatory cytokine interleukin (IL)-6, IL-8, and brain-derived neurotrophic factor were higher in the patients with rUTI than in the controls, but there were lower levels of vascular endothelial growth factor and nerve growth factor. In the patients with rUTI who recovered from acute infection, the bladders still had immature urothelium, various ultrastructural defects, and elevated urinary inflammatory cytokines. PRP injection has the potential to promote bladder recovery in some of these patients.

## 1. Introduction

Urinary tract infection (UTI) is one of the most common diseases in both urology and the local primary care clinic [1]. Although uncomplicated UTIs can usually be easily treated, recurrent UTIs (rUTIs) are common among adult women with anatomically and physiologically normal urinary tracts [2,3]. The pathogenesis of rUTI in humans involves bacterial factors and deficiencies in host defense [4]. The fundamental host defense is the antibacterial adherence mechanism in the bladder urothelium [5]. We previously reported decreased E-cadherin and increased apoptotic cells in the bladder urothelium in patients with rUTI [6], providing evidence to support a defect in the bladder defense mechanism in human rUTI. Because rUTI might be associated with an unhealed bladder after UTI episodes, intravesical treatment to promote bladder regeneration should be a reasonable option for patients with rUTI [7]. Platelet-rich plasma (PRP) injection has been used to treat inflammatory diseases in clinical practice [8,9], and a preclinical study also provided evidence to show that the hepatocyte growth factor in PRP mediated the anti-inflammatory effect [10]. Our previous study also revealed decreased urinary inflammatory cytokine after intravesical PRP injection in patients with interstitial cystitis/bladder pain syndrome [11]. We previously conducted a clinical trial to treat patients with rUTI using repeated intravesical PRP injection [12] and found that 63.6% of patients had fewer than two episodes of rUTI in the following year. Immunochemical staining revealed an increased expression of CD34 and CK20 in the bladders that underwent intravesical PRP injection, indicating recovery of urothelium maturation after treatment.

Among the bacterial factors, invasion of *Escherichia coli* in urothelial cells with the formation of intracellular bacterial communities (IBCs) plays an important role in rUTIs. Animal studies have demonstrated that IBCs are undetected by standard urine cultures, and they can evade host defense mechanisms and may persist despite antibiotic therapy [13,14]. Human bladder studies investigating rUTI are limited. A previous study discovered exfoliated IBCs in the urine of women with acute cystitis [15]; however, direct evidence for the presence of IBCs in the bladders of patients with rUTIs is still lacking. Uroplakins (UPKs) form urothelial plaques on the surface of each urothelial cell and are important in maintaining the urothelial defense mechanism [16]. Animal studies revealed that UPKs mediate the bacterial signaling pathway in UTI [16], but the expression of UPKs in human bladders with rUTI has not yet been reported. Our previous study showed lower urinary nerve growth factor (NGF) in female UTI patients with recurrence [17], suggesting that urinary cytokines might be useful in evaluating the health of the bladder after UTI. The aim of the current study is to comprehensively investigate the pathological abnormalities in the bladder of patients with rUTI who have undergone intravesical PRP injections, including ultrastructural defects in electron microscopy (EM), UPK expression, and level of urinary cytokines. The effect of intravesical PRP injection on the bladders of patients with rUTI was also investigated.

## 2. Materials and Methods

We enrolled patients with rUTI who were admitted for intravesical PRP injection between 2017 and 2021. rUTI was defined as two or more episodes of bacterial cystitis in the past 6 months or three or more infections within the preceding 12 months. [18] Only female rUTI patients without functional, structural, or pre-existing metabolic abnormalities of the urinary tract were enrolled. All of the patients had a recurrence of uncomplicated UTIs. All patients had received antibiotic treatment and clinically recovered from UTI for at least two weeks. Urinalysis and standard culture were performed for all patients. Only patients with urinary white blood cell count < 5/HPF and negative urine culture were enrolled for further assessment. Each patient underwent a comprehensive medical history investigation and urodynamic study to rule out anatomical or functional voiding dysfunction. Patients with spinal neurogenic voiding dysfunction, bladder outlet obstruction, vesicoureteral reflux, bladder stone, and bladder tumor were excluded. Eligible patients were admitted for repeated intravesical PRP injections under general anesthesia once a month for 4 months. The PRP preparation and injection procedures were reported in our previous study. [12] In brief, 50 mL whole blood was obtained from patients and centrifuged with a soft spin at 1500 rpm and a hard spin at 4000 rpm. A final 5 mL of PRP was collected. The study was approved by the ethics committee of the hospital (institutional review board no. 106-173-A). All patients were informed about the study rationale and procedures and provided written informed consent before the treatment. For comparison, women with stress urinary incontinence undergoing anti-incontinence surgery donated their bladder tissue and urine for investigation as control subjects. Patients were followed monthly for 1 year after the fourth PRP injection. Patients who had three or more episodes of UTI recurrence in the following year or two or more episodes of UTI recurrence in the following six months were considered to have an unsuccessful outcome, and the other patients (≤2 UTIs in one year and ≤1 UTI in six months) were considered to have a successful outcome.

### 2.1. Urine Sample Collection and Cytokine Analysis

Patients were asked to provide urine before the first PRP injection and 1 month after the fourth PRP injection. Urine was self-voided when patients or controls reported a full bladder sensation. The urine sample storage and centrifuge procedures were reported in our previous study [19]. We used the commercially available Milliplex Human cytokine/chemokine magnetic bead-based panel kit (Millipore, Darmstadt, Germany) to analyze urinary cytokines according to the manufacturer’s instructions. The interested targets included inflammatory cytokines such as interleukin-2 (IL-2), IL-6, and IL-8 and growth factors such as vascular endothelial growth factor (VEGF), epidermal growth factor (EGF), NGF, and brain-derived neurotrophic factor (BDNF).

### 2.2. Bladder Biopsies, Western Blotting, and Transmission EM

Patients were asked to provide bladder specimens with endoscopic biopsy at the first and fourth PRP injection (1 month after the third injection). The cold-cup biopsy procedure was identical to that of our previous study. [12] The bladder specimens were sent to our laboratory department for Western blotting and transmission EM (TEM) analysis. The Western blotting procedure has been previously reported. [20] In the current study, UP3 expression was evaluated with primary antibody (abcam, ab157801, 1:500), and GAPDH expression was used for quantification (GeneTex, GTX100118, 1:10,000). For the TEM, all bladder specimens were investigated using a Hitachi H-7500 TEM. The procedure of specimen handling for the TEM was identical to that of our previous study. [21] Bladder ultrastructure defects were graded with a three-point scale (normal, mildly, and severely defective) for the epithelium layer defect, tight junction defect, and lymphocyte infiltration in the lamina propria (by Dr. Hann-Chen Ho, who was blinded to clinical data; a representative image is shown in Figure 1). In addition, the integrity of the urothelial umbrella cells was graded as normal (umbrella cells covering 75–100% of the area of the urothelium), mildly defective (covering 25–75%), and severely defective (covering only 0–25%). Lysed organelles in the urothelium cells were graded as normal (lysed organelles presented in <25% of all epithelial cells), mildly defective (25–75%), or severely defective (>75%).

### 2.3. Statistical Analysis

The grading results in the EM findings for the patients with rUTI at baseline and after PRP injection were compared using McNemar’s test. Quantification of Western blotting and results of urinary cytokine expression in the different groups were compared with a nonparametric independent *t*-test (Mann–Whitney *U* test) or paired *t*-test (Wilcoxon signed-rank test). A *p*-value < 0.05 was considered significant. All calculations were analyzed using GraphPad Prism 8.

## 3. Results

Twenty-nine patients with rUTI (all female) completed repeated intravesical PRP injections once a month for 4 months. Table 1 lists the patients’ baseline characteristics. The mean number of episodes of UTI recurrence in the preceding year was 6.72 ± 2.71, and in the following year after the fourth injection, this number decreased to 3.45 ± 3.55. Fifteen patients (51.7%) had a successful outcome, whereas the other fourteen patients had unsuccessful outcomes. Overall, cystoscopy during the procedure did not show any significant abnormality in patients with rUTI, and 17 patients (14 successful and 3 unsuccessful outcomes) agreed to provide bladder specimens at the first and fourth PRP injections. All bladder specimens were investigated with TEM, but only 12 (10 successful and 2 unsuccessful outcomes) were qualified for further ultrastructural evaluation and grading. At baseline, mild defects of the epithelium layers, umbrella cell integrity, tight junction, lysed organelles, and lymphocyte infiltration were noted in 42–59% of the patients with rUTI (Figure 2A). Severe ultrastructural defects of the abovementioned EM characteristics were also noted in about 17–25% of the patients with rUTI. Five bladder specimens from the control subjects were also investigated with electron microscopy, and all of the five specimens did not exhibit urothelial defects. After PRP injections, although most patients still had mild urothelial ultrastructural defects, only one patient remained with a severe defect of the urothelium layer and umbrella cell integrity. Improvement in urothelial ultrastructural defects after PRP injections was noted in 25–42% of patients, whereas 33–58% of patients had a stationary bladder and 8–25% of patients experienced worsened bladder defects after PRP injections (Figure 2B). However, the bladder ultrastructure defects were not significantly changed between baseline and after PRP injection (as evaluated using McNemar’s test). Notably, IBCs in the urothelium were noted in 3 of 17 patients with rUTI before PRP injection (Figure 3A,B) but not in the bladders after PRP injection.

All 17 bladder specimens were analyzed with Western blotting to evaluate UPK3 expression. UPK3 expression was significantly higher in the control bladder specimens than in patients with rUTI at baseline (*p* = 0.0013; Figure 4A). As compared with baseline, the level of urothelial UP3 expression in patients with rUTI was not significantly changed after repeated PRP injections (*p* = 0.0543; Figure 4A). However, in the patients with rUTI who had a successful outcome of PRP treatment, the level of UPK3 expression was significantly increased as compared with baseline (*p* = 0.0068; Figure 4B).

All 29 patients provided urine samples at baseline, but only 14 patients (10 successful and 4 unsuccessful outcomes) provided urine at 1 month after the fourth PRP injection. AS compared with healthy controls, the baseline urinary inflammatory cytokine IL-6 and IL-8 levels were significantly higher in patients with rUTI, whereas the IL-2 level was significantly lower (Figure 5A). The baseline levels of urinary growth factors, including NGF and VEGF, were significantly lower in patients with rUTI than in healthy controls (Figure 5A), although the BDNF level was significantly higher. The urinary BDNF level was decreased compared with baseline, whereas the level of the other cytokine did not exhibit a significant change (Figure 5B).

## 4. Discussion

Although researchers have made a great effort to investigate the pathogenesis of rUTI, most studies have been carried out using various animal models, and the evidence from human bladders remains limited [22]. Recent research has suggested using intravesical PRP injection might have a therapeutic effect in treating lower urinary tract disorders due to regenerative deficiency [7]. We previously conducted a study to investigate the clinical efficacy of intravesical PRP injection for the patients with rUTI, and about 60% of patients had successful treatment outcomes (UTI recurrence ≤ two times in the following 12 months) [12]. In the current study, about 57% of the patients with rUTI achieved successful treatment outcomes, and the reduction in mean UTI recurrence episodes/1 year was 48%. The success rate in this study was like our previous study. We further investigate the bladder ultrastructural and urinary cytokine expression after PRP injection. Grossly, the appearance of the bladders of the patients with rUTI who recovered from acute UTI was nearly normal upon cystoscopy. Nevertheless, EM results revealed that, ultrastructurally, the bladders remained significantly denudated, especially in the uroepithelial cells. Analysis of urinary cytokines in the patients with rUTI revealed a persistently higher level of inflammatory cytokines than the control subjects. Bladder ultrastructural defects were improved in 25–42% of patients with rUTI compared with baseline, suggesting that repeat PRP injections may promote bladder healing in at least some of patients with rUTI.

Because more studies of UTI pathogenesis have focused on the acute stage and bacterial factors, the current understanding of bladder structural or functional changes after UTI recovery is still limited. A previous study in rats revealed a swollen epithelial surface and loss of epithelial cells within 2 h after acute UTI [23]. Bladder recovery could be detected 24 h after the UTI episode, and at 48 h, the bladder epithelium closely resembled that of noninfected control animals [23]. However, our study showed that patients with rUTI still had urothelial ultrastructural defects even 2 weeks after UTI recovery. The bladders of patients with rUTI were characterized by the thinning of the urothelium with remarkably loosened epithelial tight junctions and lost umbrella cells. In the remaining epithelial cells, lysed organelles could also be commonly detected. Infiltration of inflammatory cells in the lamina propria was also presented in most patients with rUTI. The bladders of patients with rUTI might remain in an unhealed urothelium and persistent inflammation status. Although a comprehensive bladder change could not be achieved using repeat intravesical PRP injections, some patients improved in urothelium defects. Animal studies revealed that the formation of the IBC plays a central role in UTI recurrence, leading to rapid replication of bacteria after the infection subsides [13,14]. A previous study revealed IBCs present in the urothelial cells collected from urine cytology [15]; our current study further provided direct EM evidence showing that IBCs were indeed present in human bladder biopsy specimens from the patients with rUTI, suggesting that bacteria could be incubated in the human uroepithelial cells. An early study used scanning EM to investigate bladder specimens from patients with chronic persisted UTI, and bacterial-persisted adherence in the urothelial cell surface was also noted [24]. The bacterial could long-term infect or colonize the bladder urothelium, and these findings support the current guideline of using long-term prophylactic antibiotics for human patients with rUTI [18].

Bacterial adhesion to the urothelium is possible through the interaction of the adhesion molecules of FimH and the subunit UPK1a of urothelial plaques, whereas UPK2 and UPK3 do not participate in this reaction [16]. Although the association between the urothelial UPK3 expression level and rUTI pathogenesis remains unclear, UPK3 is important in urothelial anti-permeability function and is considered a marker of mature umbrella cells [16,25]. Our study revealed significantly lower urothelial UPK3 expression in patients with rUTI, suggesting that the urothelium remained immature even though the gross appearance in cystoscopy was nearly normal. Although the expression of UPK3 was not significantly increased after PRP injections in all rUTI patients, those who had a successful outcome had an improvement in urothelial UPK3 expression. Our previous study revealed that the urothelial CK20, a marker for urothelial maturation, was significantly increased after the PRP injection in patients with rUTI. PRP injections might potentially promote urothelial healing and maturation in, at least, some patients with rUTI.

The urinary cytokine level has been widely investigated as a potential biomarker in noninfectious bladder benign diseases such as overactive bladder and interstitial cystitis [19,26]. Although it is unnecessary for obtaining a diagnosis of UTI, the abnormal expression of urinary cytokines may suggest a pathological change in the bladder and can potentially predict disease prognosis. The current study revealed increased levels of urinary proinflammatory cytokines IL-6 and IL-8 [27,28] in patients with rUTI at baseline, whereas the anti-inflammatory cytokine IL-2 [29] level was decreased. Despite the fact that urinary white blood cells were absent in the patients with rUTI who recovered from the acute UTI episode, persistent elevation of proinflammatory cytokines suggests that the inflammation of the bladder had not subsided completely. In the current study, the urinary inflammatory cytokines in the patients with rUTI did not decrease after PRP injection. Similarly, our previous study also revealed showed the inflammation in bladder tissue did not have significant improvement [12]. The anti-inflammatory effect of PRP injection in bladder inflammatory diseases is still uncertain. Our study also showed an abnormal level of urinary growth factors in the patients with rUTI, including higher BDNF and lower NGF and VEGF levels. Although the pathological significance is unclear, decreased urinary BDNF levels after PRP injections suggest that the bladders might be partially recovered. Our previous study also revealed increased urinary NGF levels in the patients with rUTI and first time UTI than that in control subjects, and the patients with rUTI even had a higher level of urinary NGF than the patients with first time UTI [17]. In addition, the level of urinary NGF was decreased after antibiotics treatment, but it still was higher than control at 12 weeks after the UTI episode. [17]

The main limitation of this study is the lack of a placebo control group. In addition, the bladder and urine changes might have resulted from self-recovery instead of PRP injections. However, it is hard to conduct a placebo control study for rUTI patients with intravesical injection procedures because of ethical concerns. The nature of urothelial recovery from rUTI is uncertain. The small case number and lack of quantitative analysis for the EM findings were other limitations. A different number of samples in various experiments might also have led to a bias in the analysis. Repeated bladder injection procedures could have led to bladder injury. However, most studies on rUTI have used an animal model, which might not be applicable to the disease in humans. Investigations of human bladder specimens from patients with rUTI are extremely limited, and further studies are required to explore the mechanisms of human rUTI.

## 5. Conclusions

Bladders from patients with rUTI who had recovered from acute infection still contained immature urothelium, various ultrastructural defects, and elevated urinary inflammatory cytokine levels. Laboratory evidence showed that bladder defects and maturation improved in some of the patients after intravesical PRP injections. Intravesical repeat PRP injection has the potential to reduce recurrence episodes and promote bladder recovery in patients with rUTI.

## Figures and Tables

**Figure 1 biomedicines-10-00245-f001:**
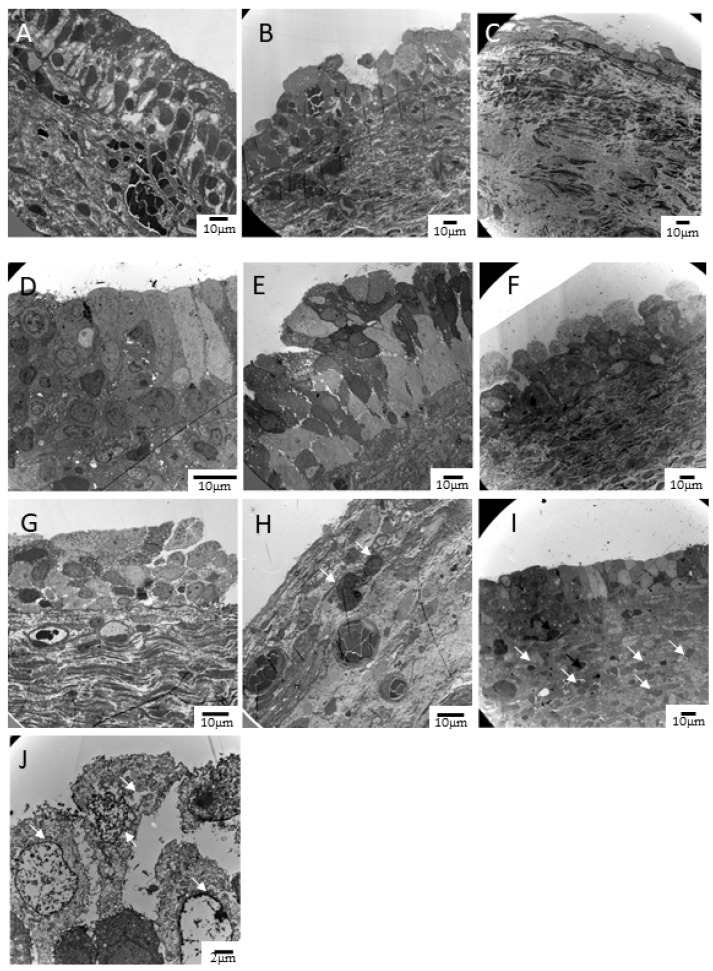
Representative figures of bladder ultrastructural defects in EM. (**A**) Normal, (**B**) mild defect, and (**C**) severe defect of the urothelial layers. (**D**) Normal, (**E**) mild defect, and (**F**) severe defect of the tight junction in the surface of urothelial cells. (**G**) Normal, (**H**) mild, and (**I**) severe infiltration of inflammatory cells (white arrow) in the lamina propria of the bladder. (**J**) Epithelial cells with lysed organelles and nucleus.

**Figure 2 biomedicines-10-00245-f002:**
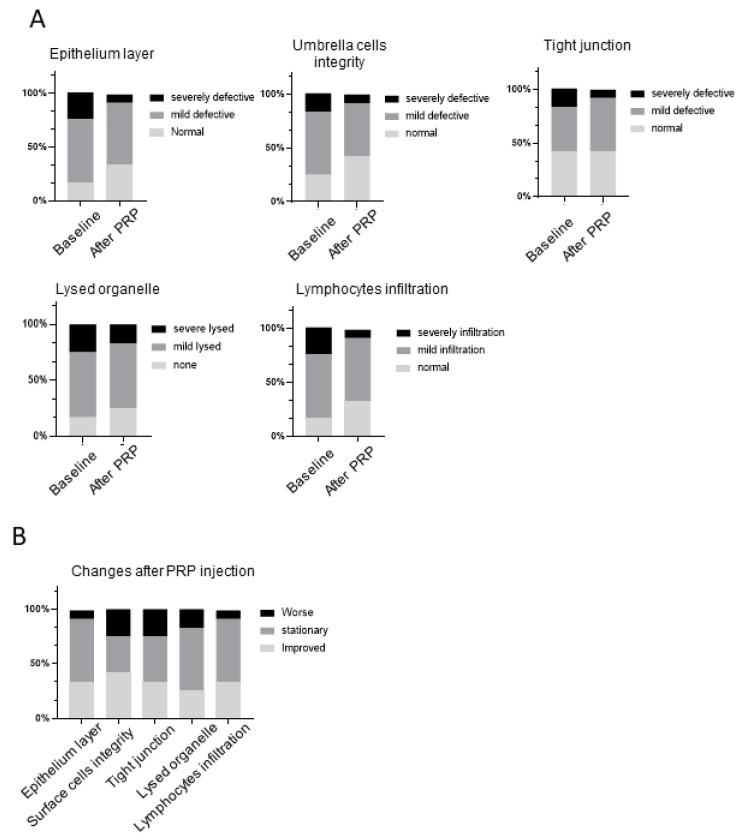
Proportion of bladder ultrastructural defects in patients with rUTI. (**A**) The bladders from patients with rUTI exhibited various ultrastructural defects on EM. (**B**) The proportion of changes in bladder ultrastructural defects in patients with rUTI after intravesical PRP injections.

**Figure 3 biomedicines-10-00245-f003:**
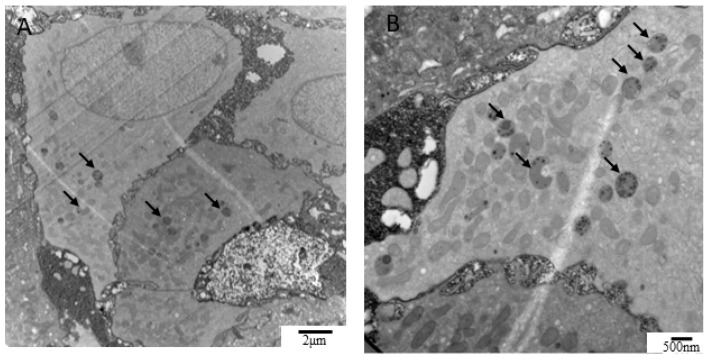
EM evidence revealed IBCs in the human rUTI bladder biopsy specimens. (**A**,**B**) IBCs were detected in the uroepithelial cells in the patients with rUTI who had recovered from an acute UTI episode, suggesting that bacterial incubation leads to UTI recurrence in patients.

**Figure 4 biomedicines-10-00245-f004:**
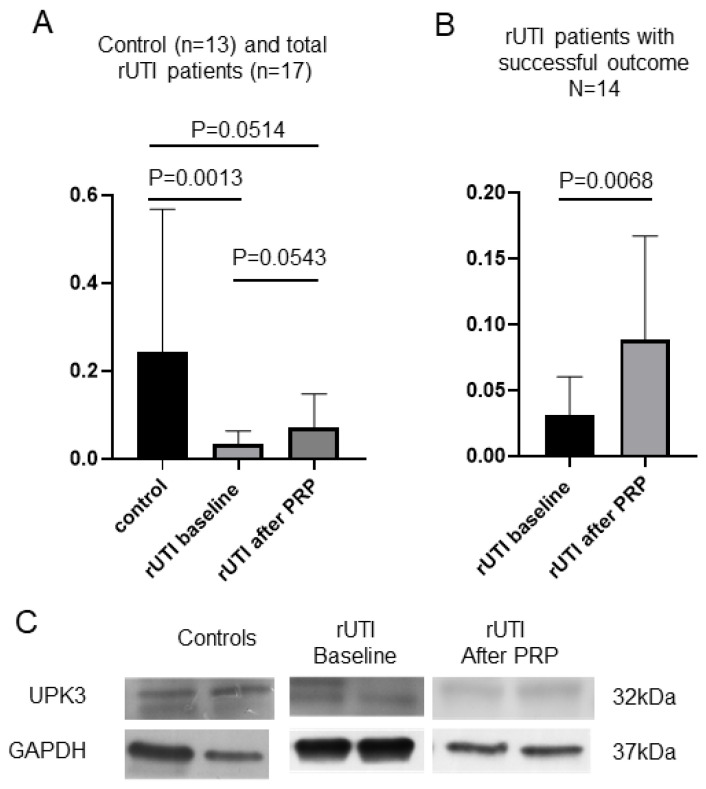
Western blot results of UPK3. (**A**) The UPK3 expression quantification result of control subjects (*n* = 13), patients with rUTI at baseline, and patients with rUTI after PRP injections (*n* = 17). The expression of UPK3 was significantly higher in the control bladder specimens than in the rUTI specimens at baseline (*p* = 0.0013) but was not significantly higher than that in rUTI specimens after PRP injections (*p* = 0.0514, nonparametric independent *t*-test). The expression of bladder UPK3 was also not significantly different between rUTI at baseline and after PRP injections (*p* = 0.0543, nonparametric paired *t*-test). (**B**) The expression of UPK3 in the patients with rUTI with successful PRP injection outcome. The expression of UPK3 was increased after PRP injection (*p* = 0.0068, nonparametric paired *t*-test). (**C**) The representative Western blotting bands from the patients with rUTI and control subjects.

**Figure 5 biomedicines-10-00245-f005:**
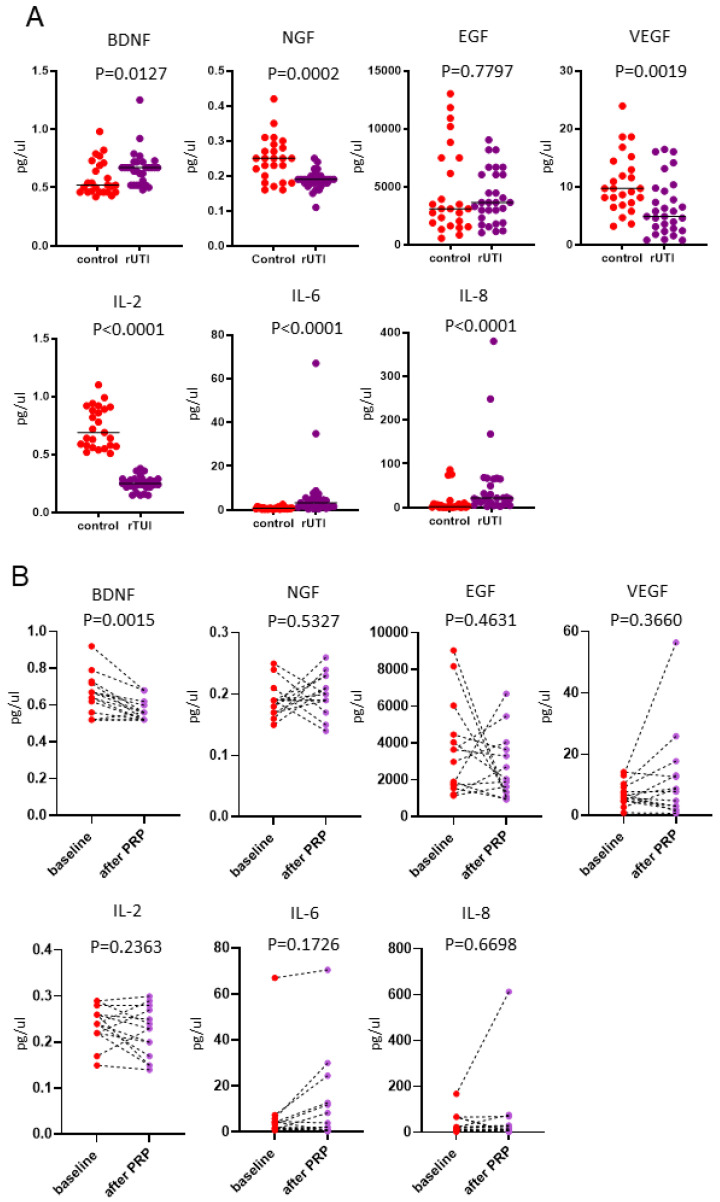
Urinary cytokine expression in the patients with rUTI and control subjects. (**A**) Urinary inflammatory cytokines and growth factors in the controls (*n* = 25) and patients with rUTI at baseline (*n* = 29). The level of urinary inflammatory cytokines IL-6 and IL-8, and BDNF levels were significantly higher in patients with rUTI at baseline (nonparametric independent *t*-test), while the levels of IL-2, NGF, and VEGF were significantly lower. (**B**) Comparison of urinary inflammatory cytokines and growth factors between the patients with rUTI at baseline and after PRP injections (*n* = 14). The level of urinary BDNF in the patients with rUTI decreased after PRP injection (nonparametric paired *t*-test).

**Table 1 biomedicines-10-00245-t001:** The baseline clinical and urodynamic characteristics in the patients with rUTIs (*n* = 29).

Age	67.34 ± 8.26
1 year UTI episode before PRP	6.72 ± 2.71
1 year UTI episode after PRP	3.45 ± 3.55
Successful outcome	15(57%)
Unsuccessful outcome	14(43%)
FSF (mL)	139.48 ± 70.47
US (mL)	239.08 ± 91.05
Qmax (mL/s)	14.07 ± 7.75
Volume (mL)	241.11 ± 142.48
PVR (mL)	73.90 ± 124.60
Pdet (cmH_2_O)	18.62 ± 12.42
CBC (mL)	316.56 ± 168.14
voiding efficiency	0.74 ± 0.34

FSF: first sensation of filling; US: urge sensation; Qmax: maximal flow rate; Volume: voided volume; PVR: post-voiding residual volume; Pdet: detrusor pressure during voiding; CBC: cystometric bladder capacity; voiding efficiency: voided volume/(voided volume + PVR).

## Data Availability

Data are available by contacting the corresponding authors.

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
