# Peer review of "Bladder Ultrastructure and Urinary Cytokine Abnormality in Patients with Recurrent Urinary Tract Infection and the Changes after Intravesical Platelet-Rich Plasma Injections"

_biomedicines, 2022, doi:10.3390/biomedicines10020245_

Round 1

Reviewer 1 Report

The authors describe the application of PRP injections into the bladder wall, with a success rate of 57%. However, I have two major concerns.

First, I would like to know the pathogenesis of the intravesical application of platelet-rich plasma to the reduction of inflammatory response. I was trying to find similar references made by the authors but I was not successful. Please, we should know the background of such an application. It should be incorporated in the introduction, and properly cited.

Second, the authors have an article in Luts (https://doi.org/10.1111/luts.12364) which has the same methodology. Please in discussion comment on the differences between this article and the article in review. Additionally, they have a control group of 17 patients in the previous article, but this control group is not mentioned in the article in review. Please comment on this.

The authors mention these lines "The main limitation of this study is the lack of a placebo control group. In addition, the bladder and urine changes might have resulted from self-recovery instead of PRP injections." I am well aware that is the main limitation of the study. Please have additional comments on this topic.

The English style should be considerably improved. 

Author Response

The authors describe the application of PRP injections into the bladder wall, with a success rate of 57%. However, I have two major concerns.

First, I would like to know the pathogenesis of the intravesical application of platelet-rich plasma to the reduction of inflammatory response. I was trying to find similar references made by the authors but I was not successful. Please, we should know the background of such an application. It should be incorporated in the introduction, and properly cited.

Reply: Thanks for your comment. Platelet-rich plasma (PRP) injection has been used to treat inflammatory diseases in clinical practice[1,2], and preclinical study also provided evidence to show the hepatocyte growth factor in PRP mediated the anti-inflammatory effect[3]. Our study also revealed decreased urinary inflammatory cytokine after intravesical PRP injection in the patients with interstitial cystitis/bladder pain syndrome[4]. We had added above information into introduction section. (page 3, line 13 to line 18)

In current study, the urinary inflammatory cytokines were higher in the patients with rUTI than the controls, but they did not decrease after PRP injection. Our previous study also revealed showed the bladder inflammation did not have significant improvement after intravesical PRP injection for the patients with rUTI[5]. The anti-inflammatory effect of PRP injection in bladder inflammatory diseases is still uncertain. We also added above sentence into the discussion section. (page 18, line 3 to line 7)

Second, the authors have an article in Luts (https://doi.org/10.1111/luts.12364) which has the same methodology. Please in discussion comment on the differences between this article and the article in review. Additionally, they have a control group of 17 patients in the previous article, but this control group is not mentioned in the article in review. Please comment on this.

Reply: Thanks for your comment. We previous conduct a study to investigate the clinical efficacy of intravesical PRP injection for the patients with rUTI, and about 60% patients had successful treatment outcome (UTI recurrence ≤two times in the following 12 months). In current study, the about 57% of the patients with rUTI achieved a successful treatment outcome, and the reduction in mean UTI recurrence episodes/1 year was 48%. The successful rate was similar to our previous study. We further investigate the bladder ultrastructural and urinary cytokine expression after PRP injection. Our previous study revealed that the urothelial CK20, a maker for urothelial maturation, was significantly increased after the PRP injection in the patients with rUTI. We added above discussion into different section in this manuscript. (page 16, line 3 to line 10; and page 17, line 26 to 27)

Current study also enrolled control subjects for urine and also bladder specimens. All the control subjects were also patients with stress urinary incontinence (page 4, line 29 to 31). Five bladder specimens from the control subjects also investigated with electron microscopy, and all of the 5 specimens did not exhibit urothelial defects. (page 8, line 19 to line 21). The UPK3 expression was investigated in 13 control subjects, and the UP3 expression was higher in the control. (page 12, line 5 to 6) Urine sample was provided by 25 control subjects, and current study revealed the level of urinary inflammatory cytokine and growth factors were significantly different between control and rUTI patients at baseline. (page 13, line 19 to page 14, line 5)

The authors mention these lines "The main limitation of this study is the lack of a placebo control group. In addition, the bladder and urine changes might have resulted from self-recovery instead of PRP injections." I am well aware that is the main limitation of the study. Please have additional comments on this topic.

Reply: Thanks for your comment. Indeed, the lack placebo control is the main limitation of this study. The time course of urothelial recovery from UTI in human has never been investigated, but previous animal study revealed the bladder recovery could be detected in 24 hours after the UTI episode, and at 48 hours, the bladder epithelium closely resembled that of noninfected control. (page 16, line 27 to 30) It is hard to conduct a placebo control study for the rUTI patients with intravesical injection procedure because of ethical concerns. The nature urothelial recovery from rUTI is uncertain. (page 18, line 19 to 21)

The English style should be considerably improved.

Reply: Thanks for your suggestion. Our manuscript had underwent editing by for English language, grammar, punctuation, and spelling by Enago, an editing brand of Crimson Interactive Inc. The certification for the English editing is available at https://doi.org/10.6084/m9.figshare.17912294.v1 . Because the revision deadline for this manuscript is only 10 days, we did not send this manuscript for editing again. We hope your understanding.

  1. Badsha, H.; Harifi, G.; Murrell, W.D. Platelet Rich Plasma for Treatment of Rheumatoid Arthritis: Case Series and Review of Literature. Case Rep Rheumatol 2020, 2020, 8761485, doi:10.1155/2020/8761485.
  2. Huang, G.; Hua, S.; Yang, T.; Ma, J.; Yu, W.; Chen, X. Platelet-rich plasma shows beneficial effects for patients with knee osteoarthritis by suppressing inflammatory factors. Exp Ther Med 2018, 15, 3096-3102, doi:10.3892/etm.2018.5794.
  3. Zhang, J.; Middleton, K.K.; Fu, F.H.; Im, H.J.; Wang, J.H. HGF mediates the anti-inflammatory effects of PRP on injured tendons. PLoS One 2013, 8, e67303, doi:10.1371/journal.pone.0067303.
  4. Jiang, Y.H.; Kuo, Y.C.; Jhang, J.F.; Lee, C.L.; Hsu, Y.H.; Ho, H.C.; Kuo, H.C. Repeated intravesical injections of platelet-rich plasma improve symptoms and alter urinary functional proteins in patients with refractory interstitial cystitis. Sci Rep 2020, 10, 15218, doi:10.1038/s41598-020-72292-0.
  5. Jiang, Y.H.; Jhang, J.F.; Hsu, Y.H.; Ho, H.C.; Lin, T.Y.; Birder, L.A.; Kuo, H.C. Urothelial health after platelet-rich plasma injection in intractable recurrent urinary tract infection: Improved cell proliferation, cytoskeleton, and barrier function protein expression. Low Urin Tract Symptoms 2021, 13, 271-278, doi:10.1111/luts.12364.

Reviewer 2 Report

The manuscript submitted by Jia-Fong Jhang et al., entitled "Bladder Ultrastructure and Urinary Cytokine Abnormality in Patients with Recurrent Urinary Tract Infection and the Changes after Intravesical Platelet-Rich Plasma Injections" is interesting. The authors present their results regarding Intravesical Platelet-Rich Plasma Injections to treat recurrent UTIs. This article continues the exploration of PRP injections in urinary bladder pathology performed by the same study group. 

The authors should try to use the journal template and include line numbers.

Here are my comments:

  1. The first line of the abstract - hard to understand
  2. "reference numbers should be placed in square brackets [ ], and placed before the punctuation;" https://www.mdpi.com/journal/biomedicines/instructions 
  3. Results - only one patient remained with a severe defect of...
  4. Table 1 it should be UTIs
  5. Please include the following manuscript in the Introduction and Discussion sections.
    Ke QS, Jhang JF, Lin TY, et al. Therapeutic potential of intravesical injections of platelet-rich plasma in the treatment of lower urinary tract disorders due to regenerative deficiency. Ci Ji Yi Xue Za Zhi. 2019;31(3):135-143. doi:10.4103/tcmj.tcmj_92_19
  6. Discussions: "Although researchers have made a great effort to investigate the pathogenesis of rUTI, most studies have been carried out using various animal models, and the evidence from human bladders remains limited" - needs some references.
  7. Animal studies revealed that the formation of the IBC plays a central role in UTI recurrence, which could lead to rapid replication of bacteria after the infection subsides. - needs a reference
  8. The Discussion section needs to be revised. The authors should try to elaborate more about the role of Bladder Ultrastructure and Urinary Cytokine Abnormality in Patients with Recurrent Urinary Tract Infections.
  9. Materials and Methods should be placed before Results
  10. Materials and Methods - please specify if there are uncomplicated UTIs or complicated UTIs. Did the authors enroll only female patients?
  11. For a successful outcome - The two episodes of UTIs should be more in more than six months to be successful.
  12. Figure 5 in the text it is J, not K.
  13. Please include a list of abbreviations used in alphabetical order.

Author Response

Dear Reviewer:

Thanks for your comments and help for this manuscript. The following is our response to your comments and questions.

The manuscript submitted by Jia-Fong Jhang et al., entitled "Bladder Ultrastructure and Urinary Cytokine Abnormality in Patients with Recurrent Urinary Tract Infection and the Changes after Intravesical Platelet-Rich Plasma Injections" is interesting. The authors present their results regarding Intravesical Platelet-Rich Plasma Injections to treat recurrent UTIs. This article continues the exploration of PRP injections in urinary bladder pathology performed by the same study group. 

The authors should try to use the journal template and include line numbers.

Here are my comments:

  1. The first line of the abstract - hard to understand

Reply: Thanks for your comment. We had revised this sentence. (page 2, line 2 to 4)

  1. "reference numbers should be placed in square brackets [ ], and placed before the punctuation;" https://www.mdpi.com/journal/biomedicines/instructions 

Reply: Thanks for your comment. We had revised all reference number.

  1. Results - only one patient remained with a severe defect of...

Reply: Thanks for the reminder, we had correct the sentence. (page 8, line 22)

  1. Table 1 it should be UTIs

Reply: Thanks for the reminder, we had correct the table title. (page 9, line 8 )

  1. Please include the following manuscript in the Introduction and Discussion sections.
    Ke QS, Jhang JF, Lin TY, et al. Therapeutic potential of intravesical injections of platelet-rich plasma in the treatment of lower urinary tract disorders due to regenerative deficiency. Ci Ji Yi Xue Za Zhi. 2019;31(3):135-143. doi:10.4103/tcmj.tcmj_92_19

Reply: Thanks for your suggestion. We had added above mentioned reference into discussion and introduction. (page 3, line 11 to 13; and page 16, line 1 to 3)

  1. Discussions: "Although researchers have made a great effort to investigate the pathogenesis of rUTI, most studies have been carried out using various animal models, and the evidence from human bladders remains limited" - needs some references.

Reply: Thanks for the reminder. We had added the reference. (page 16, line 1)

  1. Animal studies revealed that the formation of the IBC plays a central role in UTI recurrence, which could lead to rapid replication of bacteria after the infection subsides. - needs a reference

Reply: Thanks for the reminder. We had added the reference. (page 17, line 4)

  1. The Discussion section needs to be revised. The authors should try to elaborate more about the role of Bladder Ultrastructure and Urinary Cytokine Abnormality in Patients with Recurrent Urinary Tract Infections.

Reply: Thanks for the comment. We had added and revised some discussion about bladder ultrastructural and urinary cytokine in the patients with UTI. (page 16, line 1 to 10; page 17, line 8 to 13; page 17, line 25 to 27; page 18, line 3 to 7; and page 18, line 11 to 15; page 18, line 19 to 21)

  1. Materials and Methods should be placed before Results

Reply: Thanks for the reminder. We had revised the manuscript arrangement.

  1. Materials and Methods - please specify if there are uncomplicated UTIs or complicated UTIs. Did the authors enroll only female patients?

Reply: Thanks for your comment. We enrolled only female rUTI patients without pre-existing metabolic, functional, or structural abnormalities of the urinary tract. The patients are recurrence of uncomplicated UTIs. We added the clarification in the method section. (page, 4, line 12 to line 14 )

  1. For a successful outcome - The two episodes of UTIs should be more in more than six months to be successful.

Reply: Thanks for the reminder. We had added the description in the method section. (page 4, line 35 to line 35)

  1. Figure 5 in the text it is J, not K.

Reply: Thanks for the reminder. We had revised the figure legend.

  1. Please include a list of abbreviations used in alphabetical order.

Reply: Thanks for your suggestion. We added a list of abbreviations to the end of the manuscript.

Round 2

Reviewer 1 Report

The authors significantly improved thier manusript and added new information about the subject. Therefore, I have no further comments. 

Author Response

The authors significantly improved thier manusript and added new information about the subject. Therefore, I have no further comments. 

Reply: Thanks for your help with this manuscript again. Your comment helped us to improve our manuscript a lot. We also checked and corrected the typos and grammar errors in this manuscript.

Reviewer 2 Report

Dear Authors,

First of all, I want to congratulate you on your work. The manuscript is improved.

The manuscript needs English revision before it can be processed further.

Author Response

Dear Authors,

First of all, I want to congratulate you on your work. The manuscript is improved.

The manuscript needs English revision before it can be processed further.

Reply: Thanks for your help with this manuscript. Your comment helped us to improve our manuscript a lot. Initially, our manuscript had undergone editing for the English language by Enago, an editing brand of Crimson Interactive Inc. The certification for the English editing is available at https://doi.org/10.6084/m9.figshare.17912294.v1 We also checked and corrected the typos and grammar errors for the last revision of this manuscript with the online English correcting website Grammarly.com. 

Thanks for your help again. 
